# Preparation Method of Lunar Soil Simulant and Experimental Verification of the Performance of an Impact Penetrator for Lunar Soil Exploration

He Li [1,2] , Yuanbo Li [1], Minyu Wei [1] and Yi Shen [2,3,*]

1    College of Mechanical and Electronic Engineering, Shandong University of Science and Technology, Qingdao 266590, China; lihe2018@sdust.edu.cn (H.L.); liyb_en@sdust.edu.cn (Y.L.); weimy2021@sdust.edu.cn (M.W.)
2    State Key Laboratory of Robotics and System, Harbin Institute of Technology, No. 92, Xidazhi Street, Nangang District, Harbin 150001, China
3    KUKA China, No. 3, Liaoxin Road, Shunde District, Foshan 528311, China
*    Correspondence: yi.shen@kuka.com

**Abstract:** The exploration and investigation of lunar soil can provide necessary information for human beings to understand the Moon's geological evolution history and solar activity, and is also of great significance for human beings to search for new energy sources. The impact penetrator can dive to a certain depth below the lunar surface, depending on soil compaction effect, and obtain lunar soil detection data by using the onboard sensors. The penetrator has the advantages of small size, light weight, low power consumption and long-term detection ability. In order to verify the diving performance of the developed impact penetrator, a great deal of lunar soil simulant, with physical and mechanical properties similar to a real lunar soil sample, was prepared, which lay the foundation for experimental research. Experiments on the influences of mass–stiffness parameters and dynamic parameters were conducted to obtain reasonable parameter-matching effects and driving parameters. The penetrating experiments in lunar soil simulant, with different relative compaction parameters, indicated that the penetrator could penetrate the simulated lunar soil with high relative compaction, and the penetration depth could reach to 545 mm after 894 shocks in lunar soil, with a relative compaction of 85%. This study on the impact penetrator can provide a feasible approach for in-situ exploration of lunar soil.

**Keywords:** lunar exploration; impact penetrator; lunar soil simulant; penetrating depth

## 1. Introduction

The exploration of the geological composition, geological evolution, and physical and chemical properties of lunar soil profile helps us to have a deeper understanding of the Moon [1,2]. Conventional in-situ exploration technologies for landing missions mainly include drilling, shoveling, penetrating, tunneling and other sampling methods to obtain surface and subsurface samples of lunar soil profile [3–5]. In the detection tasks, the penetrating detection methods can obtain soil mineral categories, and soil intrinsic features, such as mechanical properties, thermal properties and electrical properties. They can also acquire subsurface soil profile information, such as particle size distribution, density, water content, heat flow, seismic wave and other vertical distribution characteristics. Penetrating detection has great scientific and engineering significance for lunar exploration [6].

Compared with the drilling method, the penetrating method is more advantageous for in-situ exploration. The probers penetrate into the soil body, depending on in-situ soil compaction, with less disturbance, and better profile fitting characteristics. Moreover, accurate in-situ data can be obtained easily. In order to obtain more abundant longitudinal detection data, many countries are working on increasing the penetrating depth of the

probe. However, the general rigid penetrating detectors need more driving power and longer body size in order to obtain greater detection depth [7–9].

The operating mode of penetration detectors can be divided into static pressure and dynamic impact types. Static pressure type penetrators generally act on shallow soil. Due to the stable and measurable force, the penetrating process of this kind of penetrator is simple. They can measure the soil internal friction angle, cohesion, density and other mechanical parameters [10,11]. Dynamic impact type penetrators can produce large impact force instantaneously and can penetrate into deeper subsurface soil. Therefore, they are suitable for sensing detection at a large depth [12,13].

According to the arrangement of the power, impact penetrators mainly include penetrators with external power and penetrators with internal power. A typical representative of penetrators with external power is the Self-Recording Penetrometer used in Apollo missions. It is used to test the mechanical properties of lunar soil. The diameter of the penetrator cone is 12.8 mm and the maximum penetrating depth is 74 cm [14]. Rosetta, launched in 2004, successfully reached the comet called 67 p/Churyumov-Gerasimenko in 2014 [15,16]. The MUPUS impact-penetrating probe, which is mounted on the guide rail at the end of the Philae lander arm, is also a kind of penetrator with external power. It penetrates downward continuously through the impact of an electromagnetic hammer on the body of the penetrator. According to the relationship between the energy dissipation during hammering and the diving depth, the mechanical properties of planet soil can be deduced [17]. The MUPUS PEN, a thermal detector, is attached to the wall of the probe's penetration rod to detect changes of ambient temperature. Honey Bee designed a penetrator with external power for loose lunar soil. The air passage of the penetrator runs through the body and extends to the side of the head. The momentum exchange between gas and lunar soil is used to transport lunar soil to the lunar surface, and the lunar soil is collected and discharged by the collection mechanism. During penetration, heat flow profile sensors can be arranged. When diving to a predetermined location or encountering a rock, the penetrator can be pulled back by the connected cables and dive at a different point. This penetrating method has the advantages of high efficiency and small volume [18,19].

Impact penetrators with internal power mainly include KRET, developed by the Space Technology Research Center of the Polish Academy of Sciences, the Moon Mars Underground Mole (MMUM), designed by Ames Research Center, HP3 on NASA's InSight Mars Lander of the USA, and IMS developed by the German Space Agency. KRET uses a screw nut pair driven by a motor to realize the compression and release of hammer mechanism. The release and connection actions are completed by a spring lock. KRET's maximum penetrating depth is 1.85 m [20]. A motor in the MMUM drives the spring compression through a rotating shaft. When the unlocking point is triggered, the compression spring is released and the connected impact mass is driven to act on the penetrator head. It can impact 12 times per minute, and the acting force exceeds 63 N, which is enough to achieve the maximum diving depth of 2 m. The penetrator head has a sampling function and can collect 7 g soil samples. The temperature sensor can measure the thermal properties of the soil and the Raman spectrometer can detect the mineral composition of the soil. After diving to a certain depth, equipment on the planet's surface can use reels and tethers to lift the penetrator to the surface [21]. HP3 has a penetrating depth of 5 m, a mass of 3 kg and a peak power of 2 W. The penetrator is equipped with a seismograph, radio tracker and heat flow probe, which can measure parameters, such as temperature, penetrating depth, inclination angle, heat conductivity and soil density etc. [22]. The impact penetrator IMS developed by the German Space Agency separates the diving actuator from the scientific payload. Its penetrating depth is 2 m [23].

Harbin Institute of Technology took the lead in China on the development of electromagnetic actuated penetrators for lunar soil profile detection, and initiated the impact penetrating detection of other extraterrestrial objects [24,25]. Since 2009, the team has carried out plenty of pre-research work on the impact penetrator, especially in the period when it undertook the third stage tasks for lunar sampling in the National Lunar Explo-

ration Project [26,27]. Research topics on peristaltic tunneling penetrators, impact-actuated penetrators, and auger drilling and coring techniques have been carried out successively. A variety of prototypes of penetrators have been developed to verify the feasibility of the principle of impact-actuated penetrating methods [28,29].

The penetrator with energy storage method based on a rope-driven cam mechanism is a kind of impact penetrator with great potential application prospects. In the early stage, the optimal design parameters of the penetrator were obtained by theoretical analysis and simulation, such as the configuration of the penetrator, work procedures, and the driving parameter range in the penetrating process. In order to verify the feasibility of the impact penetrator for lunar surface exploration in the future, it is necessary to test the penetrating efficiency of the prototype. In this paper, the preparation method of simulated lunar soil samples for experiments was proposed to provide uniform basic conditions for conducting decoupling and comprehensive verification experiments. Then, the principal prototype with optimization parameters, and the auxiliary experimental devices, were developed, and penetrating experiments were carried out to verify the performance of the proposed penetrator.

## 2. Overall Scheme of Impact Penetrating Detection System

For lunar subsurface exploration, the concept of in-situ lunar soil detection using the impact penetrator is proposed. The overall scheme is shown in Figure 1a. The penetrating system consists of the impact penetrator, the umbilical cable, and the auxiliary device. The impact penetrator completes the penetrating work with the assistance of the auxiliary device. First, the penetrator is driven by a motor and stores energy to an impact hammer through a specific energy storage mechanism. When the energy is stored to a certain degree, the impact hammer is suddenly released, and the impact hammer hits the penetrator body at a certain speed. Under the action of the impact force, the penetrator achieves a certain displacement along the vertical direction in the lunar soil. This process goes on periodically, and the impact penetrator will penetrate to a certain depth in the lunar subsurface. Finally, the sensors start up and the data processing system conducts data collection and analysis. The schematic diagram of working principle and energy conversion process is described by Figure 1b. The energy storage unit relies on a cam mechanism to convert the energy output by the motor into the elastic potential energy of the spring so as to store it.

When the compressed spring is released, the elastic potential energy is converted to the kinetic energy of the hammer. The hammer with great kinetic energy impacts the penetrator head with the guidance of a support guider. Finally, the impact energy is transferred to the soil around the penetrator, which will cause the destruction of the soil body. In the above process, the main factors affecting the performance of the penetrator include the actuation ability of the motor, impact energy transfer process, the structural parameters of the penetrator, physical and mechanical parameters of lunar soil, etc. Among these factors, the penetrator's structural parameters and impact energy transfer efficiency are the most important factors, on account of the fact that the physical and mechanical parameters of lunar soil and the driving power of the motor usually remain unchanged [29,30]. According to the penetrator's structure shown in Figure 1c, the article focused on the structural parameters of the penetrator, matching this with problems between the mass and spring stiffness of the core unit, dynamic parameters, etc. In order to simulate the physical and mechanical parameters of real lunar soil, the preparation methods and techniques of lunar soil samples were urgent matters to be investigated before the experimental studies.

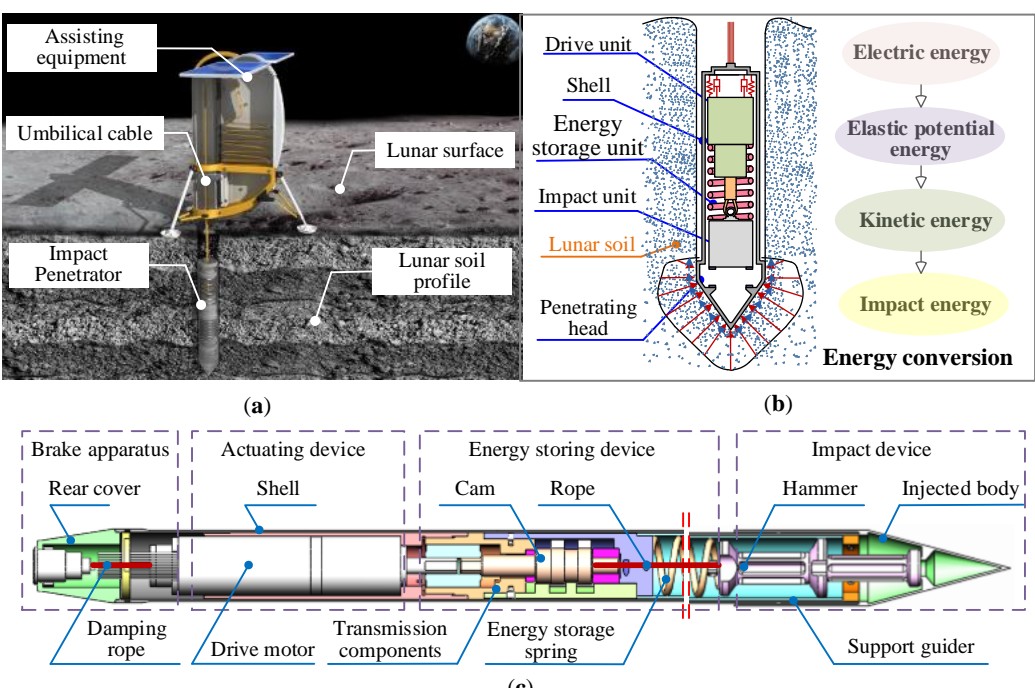

**Figure 1.** Scheme of impact penetrating detection system: (**a**) a schematic diagram of the working scenario; (**b**) schematic diagram of working principle and energy conversion process; (**c**) the three-dimensional diagram of cam-cable-driven mechanism energy-storing type penetrator prototype.

## 3. Preparation Techniques of Simulated Lunar Soil Samples

To ensure the repeatability and effectiveness of follow-up experimental research, a great deal of lunar soil simulant, with physical and mechanical properties which are similar to that of real lunar soil samples, was urgently needed. Actually, due to the limitation of current technical measures, it is not yet possible to obtain simulated lunar soil that is completely consistent with real lunar soil. Therefore, in the process of preparing the lunar soil simulant, it was advisable and necessary to select specific key indices based on all ground verification experiments. According to the analysis of stratification information of lunar soil, as well as components and characteristics of the lunar soil particle, the key physical and mechanical parameters affecting the penetrating performance of the penetrator included the following: mineral composition, density, void ratio, shear resistance and compressibility of lunar soil. Thus, these technical indicators should be taken into account in the preparation process of lunar soil simulant.

### 3.1. Analysis on Materials of Lunar Soil Simulant

3.1.1. Mineral Composition

The raw materials of the lunar soil simulant, named GUG-1B, used in this paper came from Tashan in Nanjing, Jiangsu Province. Their main ingredients and proportions in volume are shown in Table 1. They were composed of plagioclase, peridot, pyroxene, opaque mineral, volcanic glass and other components. Among them, the first three ingredients made up the majority of the raw materials. The proportion of other components in the materials was very small, and these components had little effect on the physical and mechanical parameters of the lunar soil simulant. The mineral composition of the lunar soil simulant's raw materials from Tashan was similar to that of the real lunar soil at the Apollo-14 landing site [31].

**Table 1.** The raw materials' composition and proportion in volume of GUG-1B.

| Components | Plagioclase | Peridot | Pyroxene | Opaque Mineral | Volcanic Glass | Other Components |
|---|---|---|---|---|---|---|
| Proportion (%) | 59.4 | 16.5 | 14.1 | 4.5 | 5.0 | 0.5 |

According to the particle image analyzer, the particle morphology of the simulated lunar soil was mostly angular, subangular or long strip, which was similar to actual lunar soil particles.

In terms of chemical composition, CUG-1B lunar soil simulant was generally similar to the average chemical composition of lunar soil at the Apollo14 sampling point [32]. However, compared with real lunar soil samples, the content of CaO in the lunar soil simulant was lower than that in real lunar soil, while the contents of $Na_2O$ and $K_2O$ were higher than that of real lunar soil samples. After dehumidifying and crushing the above-mentioned olive basalt materials, particles larger than 1 mm were processed and sieved in situ, and particle samples smaller than 1 mm were processed by the Raymond milling method. The particle sizes after processing were 0.1–1 mm, 0.075–0.1 mm, 0.05–0.075 mm, 0.025–0.05 mm, 0.01–0.025 mm and 0–0.01 mm. The content of each component of the lunar soil simulant with each particle size gradation is shown in Table 2.

**Table 2.** The particle grain size of lunar regolith.

| Sample No. | 1 | 2 | 3 | 4 | 5 | 6 |
|---|---|---|---|---|---|---|
| Particle's size range (mm) | 0~0.01 | 0.01~0.025 | 0.025~0.05 | 0.05~0.075 | 0.075~0.1 | 0.1~1 |
| Median size (mm) | 0.0093 | 0.01 | 0.029 | 0.064 | 0.090 | 0.41 |

Comparing the particle size of the lunar soil simulant in Table 2 with real lunar soil particle size, it can be seen that the lunar soil simulant processed and screened by the Raymond milling method covered the real lunar soil particle size range. The simulated lunar soil particle size in range of 0.1 mm–1 mm was the closest to the average particle size of real lunar soil, so the simulated lunar soil in this range was used as the raw materials in subsequent research.

3.1.2. Density of Lunar Soil Simulant

On the premise of similarity in mineral composition and chemical composition, the simulated lunar soil's density of CUG-1B was compared with real lunar soil's density, as shown in Figure 2. According to the analysis in Figure 2, the simulated lunar soil's density of CUG-1B could cover most of the distribution range of real lunar soil's density.

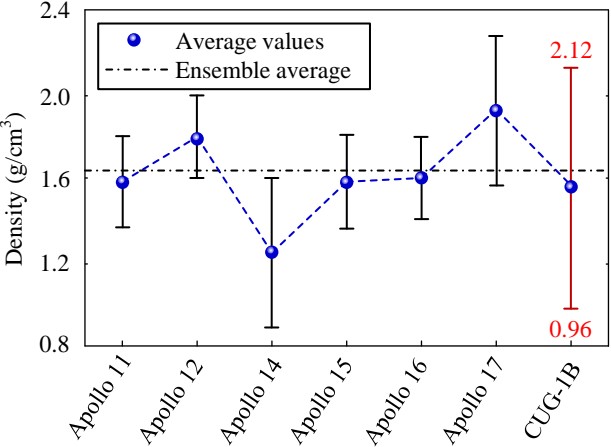

**Figure 2.** The comparison of lunar soil simulant and lunar regolith's density.

### 3.1.3. The Void Ratio of Lunar Soil Simulant

The parameters of density, void ratio and relative density under loose and compact conditions of lunar soil simulant CUG-1B and lunar regolith at the landing sites of Apollo and Luna are shown in Table 3. The maximum and minimum void ratios of CUG-1B were 1.93 and 0.884, respectively. Compared with real lunar soil, the data regarding lunar soil simulant CUG-1B were close to that of real lunar soil.

**Table 3.** The lunar soil simulant's density, relative density and void ratio.

| Sample Name | Density under Loose Condition (g/cm$^3$) | Density under Compact Condition (g/cm$^3$) | Void Ratio under Loose Condition | Void Ratio under Compact Condition |
|---|---|---|---|---|
| CUG-1B | 0.96–1.09 | 1.29–2.12 | 1.93–0.884 | 0.391–1.213 |
| Apollo 11 | 1.36 | 1.80 | 1.21 | 0.67 |
| Apollo 12 | 1.15 | 1.93 | - | - |
| Apollo 14 | 0.89 | 1.55 | 2.26 | 0.87 |
| Apollo 15 | 1.10 | 1.89 | 1.94 | 0.71 |
| Luna 16 | 1.12 | 1.79 | 1.69 | 0.67 |
| Luna 20 | 1.04 | 1.80 | 1.88 | 0.67 |

### 3.1.4. The Shear Resistance and Compressibility of Lunar Soil Simulant

The soil mechanics parameters of lunar soil simulant were measured by shear resistance experiments. The parameters of cohesion and internal friction angle of simulated lunar soil and real lunar soil samples measured by experiments are shown in Figure 3a. The internal friction angle of CUG-1B lunar soil simulant was in the range of 29.1°–34.23°, and the cohesion was in the range of 0.33–5.48 kPa. For real lunar soil, the internal friction angle ranged from 30° to 50° and the cohesion ranged from 0.03 kPa to 2.1 kPa. The variation range of internal friction angle for lunar soil simulant was within the range of real lunar soil's internal friction angle at the Apollo 17 sampling site, but the values were at a relatively low level. Under the condition of the same density, the internal friction angle of lunar soil increased roughly with the increase of particle size. In the preparation of lunar soil simulant, the integral friction angle could be improved by adding some large particle samples. According to Figure 3b, the variation range of lunar soil's cohesion was basically covered by the cohesion range of lunar soil simulant CUG-1B.

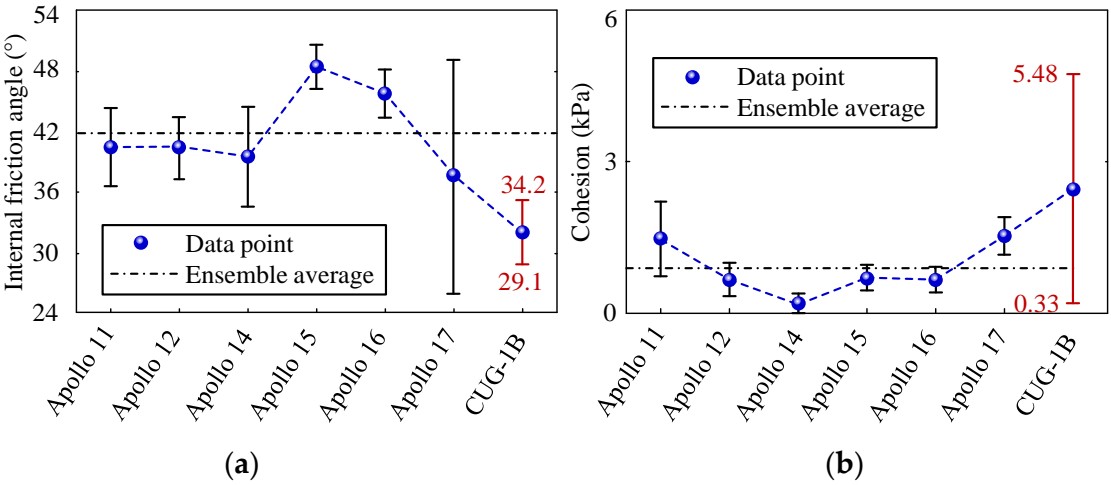

**Figure 3.** The comparison on internal friction angle and cohesion of lunar soil simulant and lunar regolith: (**a**) comparison on internal friction angle; (**b**) comparison on cohesion.

The physical and mechanical parameter comparison between lunar soil simulant and real lunar soil is shown in Table 4. Based on the above analysis, it could be found that the mineral category and chemical composition of simulated lunar soil were similar to real

lunar soil, and simulated lunar soil CUG-1B could be used as the testing materials in the subsequent ground verification experiments of the impact penetrator.

**Table 4.** Physical and mechanical parameter's comparison between lunar soil simulant and real lunar soil.

| Parameters | Real Lunar Soil | Lunar Soil Simulant | Unit |
|---|---|---|---|
| Particle size's range | <1 mm | <1 mm | mm |
| Density | 1.3~2.29 | 1.29~2.12 | g/cm$^3$ |
| Internal friction angle | 30~50° | 29.1~34.2 | ° |
| Cohesion | <0.03~2.1 | 0.33~5.48 | kPa |
| Coefficient of compressibility | <3 | 0.01~1.19 | - |

### 3.2. Preparation of Lunar Soil Simulant

The relative compaction of simulated lunar soil was the main factor to determine the penetration resistance and mechanical behavior after the raw materials of lunar soil, the configuration of the penetrator and the boundary parameters of the lunar soil barrel were determined. The relative compaction could be changed and controlled by adjusting process parameters during sample preparation. Based on the principle of equivalence and coverage on operating load, a three-dimensional vibration compaction equipment was developed suitable for CUG-1B raw materials, and large-scale lunar soil simulant was prepared for the penetrator, which would support its ground test.

Three-dimensional vibration compaction is a method to obtain simulated lunar soil samples with high relative compaction. The simulated lunar soil samples prepared by this method have good uniformity along the depth direction. Moreover, lunar soil samples with high relative compaction can be prepared with low pressure.

The method relies on vibration and pressure to press the solid powder together. The compacting force is small and the relative compaction of the prepared sample is high, which is suitable for the sample preparation of lunar soil simulant. Based on vibration compacting devices, the position relationship between the lunar soil particles was changed by vibration. Under the pressure load, the void ratio of the lunar soil particles decreased gradually, so as to improve the relative compaction of the simulated lunar soil. Figure 4 shows the working principle of the down-vibration up-pressure process. Under the action of vibration, the internal friction of the material was sharply reduced, the shear strength was reduced, and the compressive resistance became very small. Therefore, it was easily compacted under the combined action of compressive load and vibration load. In the process of preparation of the lunar soil simulant, the relative compaction and density of the simulated lunar soil were measured by the mass-volume method, and then the mass of the simulated lunar soil was recorded each time. After the vibration was completed, the height of soil body in the test tank was measured, so as to obtain the volume of the simulated lunar soil at this stage. The density of the simulated lunar soil and relative compaction at this stage were obtained.

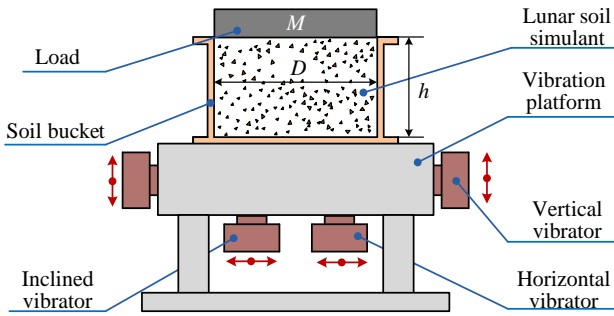

**Figure 4.** The compaction principle of the three-dimensional vibration.

The lunar soil simulant preparation device mainly included a lunar soil barrel, three-dimensional vibration platform, pressure load, etc. The performance indicators of this equipment are shown in Table 5.

**Table 5.** The performance parameters of lunar soil simulant preparation device.

| Parameters | Interior Dimensions of Lunar Soil Barrel (mm) | Number of Exciting Points | Exciting Force (kN) | Vibration Frequency (Hz) | Vibration Amplitude (mm) |
|---|---|---|---|---|---|
| Values | Diameter: 630 Height: 445 | 6 | 30 | 20~50 | 0.5~2.5 |

The preparation process was as follows: (1) The simulated lunar soil bucket was transferred to the vibration working area and fixed on the three-dimensional vibration platform. (2) A certain quantity of raw materials for simulated lunar soil, with certain particle size, was fed into the soil barrel at one time. (3) After filling, the compressive load was placed to the soil surface through the crane. (4) The three-dimensional vibration platform was started, and the device vibrated in the Z direction for 5 min at a frequency of 30 Hz. Then it vibrated in the X, Y and Z directions for 15 min at a frequency of 30 Hz. The above process was a vibration cycle. (5) The density of the simulated lunar soil was detected. When the simulated lunar soil reached the required density, the vibration of the vibration platform was stopped and the pressure load was lifted out. (6) By repeating the above loading and compaction processes, simulated lunar soil with a certain depth and density that met the penetrator's testing requirements was finally obtained.

By investigating the physical properties of real lunar soil samples and simulated lunar soil samples, it could be seen that the physical parameters, such as shear resistance and compressibility of lunar soil, corresponded to the relative compaction of lunar soil. Table 6 shows the corresponding relationship of parameters, such as density, void ratio, cohesion and internal friction angle of lunar soil simulant CUG-1B with particle size ranging from 0.1 mm to 1.0 mm under different relative compaction.

**Table 6.** The physical mechanical parameters of lunar soil simulant.

| Particle Diameter (mm) | Relative Compaction (%) | Density (g/cm$^3$) | Void Ratio | Internal Friction Angle (°) | Cohesion (kPa) |
|---|---|---|---|---|---|
| | 75 | 1.99 | 0.477 | 30.53 | 0.33 |
| | 80 | 2.02 | 0.455 | 31.42 | 0.93 |
| 0.1–1.0 | 85 | 2.05 | 0.434 | 32.33 | 1.47 |
| | 90 | 2.08 | 0.412 | 33.28 | 2.08 |
| | 95 | 2.12 | 0.391 | 34.23 | 2.72 |

## 4. Experiments

### 4.1. Mass and Stiffness Parameter-Matching Experiment

Structural components of the penetrator with cam-rope driven energy-storage type are shown in Figure 5a. It mainly included a penetrator head, support guider, impact hammer, cam-rope driven mechanism, damping rope, drive motor, buffer spring and energy storage spring, etc.

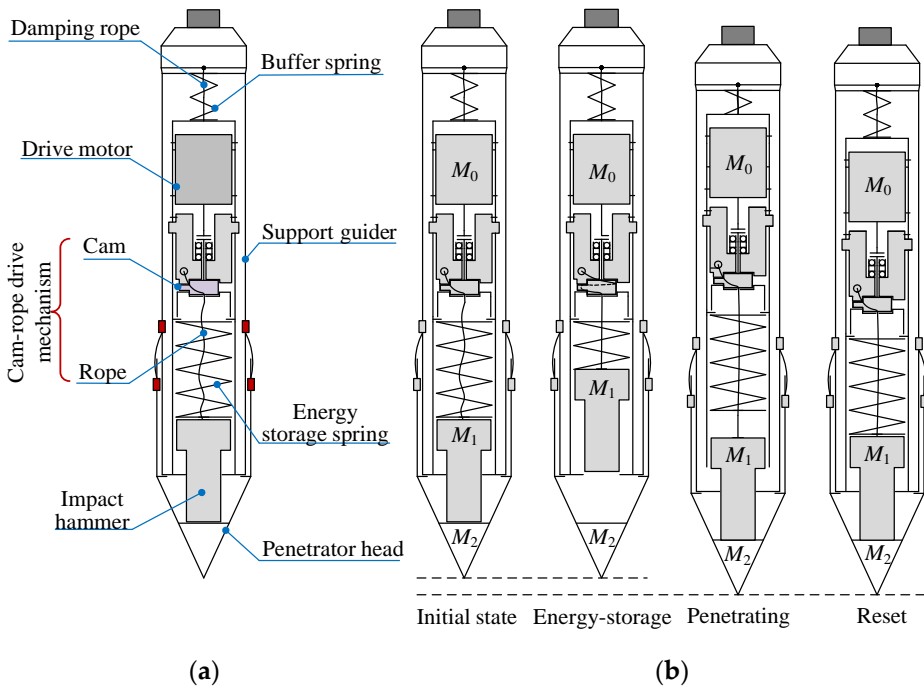

**Figure 5.** The structural components of cam-cable-driven mechanism energy-storing type penetrator and its working process: (**a**) structural components; (**b**) working process.

The penetrator's working process consisted of four stages: initial stage, energy storage stage, penetration stage, and reset stage. In the energy-storage stage, the motor drives the cam to rotate forward. When the cam rotates to 90°, the rope is in the pretension state; When the cam rotates to 180°, the rope pulls the impact hammer and compresses the energy storage spring for energy-storage. When the cam rotates to 360°, the compression of the spring reaches the maximum value, and the rope is in the critical state of release. In the penetrating stage, the rope is separated from the impact hammer, and the energy storage spring releases all the elastic potential energy. Then, the impact hammer impacts the penetrator body at a certain speed, so that a certain depth of penetration in the soil is realized. At the same time, the motor and related components rebound back and compress the buffer spring. Finally, the drive motor continues to rotate and enters the reset stage, being ready to start the next cycle. The workflow of the impact penetrator is shown in Figure 5b.

According to the above working principle and process, the matching of mass and stiffness parameters for core elements of the penetrator was an important step affecting the impact energy transmission efficiency. Considering the weightlessness on the Moon, a test platform for impact transfer characteristics was designed and built to analyze the influence of the mass and stiffness of the penetrator's core unit on the energy transfer efficiency.

As shown in Figure 6, the mass and stiffness parameter-matching testing device consisted of an impact-transferring test platform, high-speed camera, data acquisition system, and penetrator prototype. When the driving capacity of the motor was determined, the mass of the impact actuator was also determined. The initial mass of impact actuator ($M_0'$) was 289 g. The initial mass of the impact hammer ($M_1'$) was 355 g, which was determined by the envelope dimensions and its material density. Once the masses of all components of the penetrator were determined, the penetrator's initial mass was also confirmed, which was 846 g. The match mass ($M_x$) varied from 64 g to 1024 g with an incremental step of 64 g in the parameter-matching experiment. The mass of different counterweights was connected with the impact actuator, the impact hammer and the penetrator through the screw to meet the requirements of different mass-matching experiments. Buffer spring

stiffness ($K_0$) and energy storage spring stiffness ($K_1$) varied from 0.3 N/mm to 9.8 N/mm. The experimental parameters are shown in Table 7.

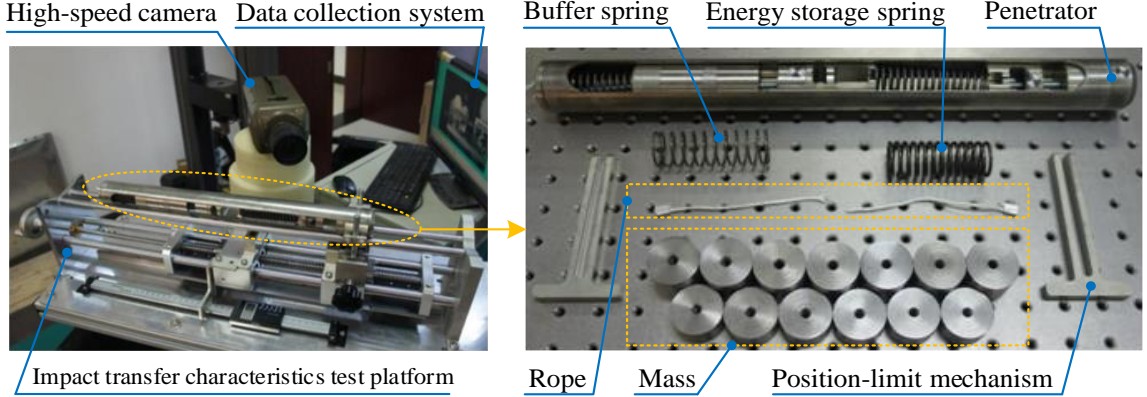

**Figure 6.** The major component of matching and stiffness matching parameters.

**Table 7.** The parameters used in the matching experiments.

| Parameters | Symbols | Values | Units |
|---|---|---|---|
| Impact actuator's envelope dimensions | - | Diameter: 33; Length: 70 | mm |
| Impact hammer's envelope dimensions | - | Diameter: 33; Length: 80 | mm |
| Penetrator's envelope dimensions | - | Diameter: 40; Length: 400 | mm |
| Impact actuator's mass | $M_0'$ | $289 + M_x$ | g |
| Impact hammer's mass | $M_1'$ | $355 + M_x$ | g |
| Penetrator's mass | $M_2'$ | $846 + M_x$ | g |
| Range of matching mass | $M_x$ | 64~1024 | g |
| Buffer spring stiffness | $K_0$ | 0.3/0.5/1/2/2.9/3.9/4.9/9.8 | N/mm |
| Energy storage spring stiffness | $K_1$ | 0.3/0.5/1/2/2.9/3.9/4.9/9.8 | N/mm |

Before the experiments, the required matching mass unit was selected and connected with the impact actuator, the impact hammer and the penetrator by the screw, and the selected buffer spring and energy storage spring were put inside the penetrator. Then, the penetrator was placed on the impact-transferring testing platform, and the energy storage spring was compressed by a limit mechanism, and the compression amount was read by digital calipers. The rope used to limit the release of the energy storage spring was installed in the limit slot between the impact actuator source and the impact hammer. The Mark points used for high-speed camera capture were attached to the impact actuator, impact hammer and penetrator. Then, the data acquisition system was started and the high-speed camera was adjusted to the best condition required for the test.

Experimental procedure: First, the high-speed camera was turned on and recording started. Then, the rope was cut and the energy storage spring was released. The impact actuator and the impact hammer moved in opposite directions under the action of the spring. The actuator moved backward and compressed the buffer spring, while the impact hammer moved forward and collided with the penetrator. After the collision, the impact hammer and the penetrator moved forward together for a certain distance and stopped under the action of resistance. Finally, the relevant test program of the data acquisition system recorded and saved the moving distance data. The experimental process is shown in Figure 7.

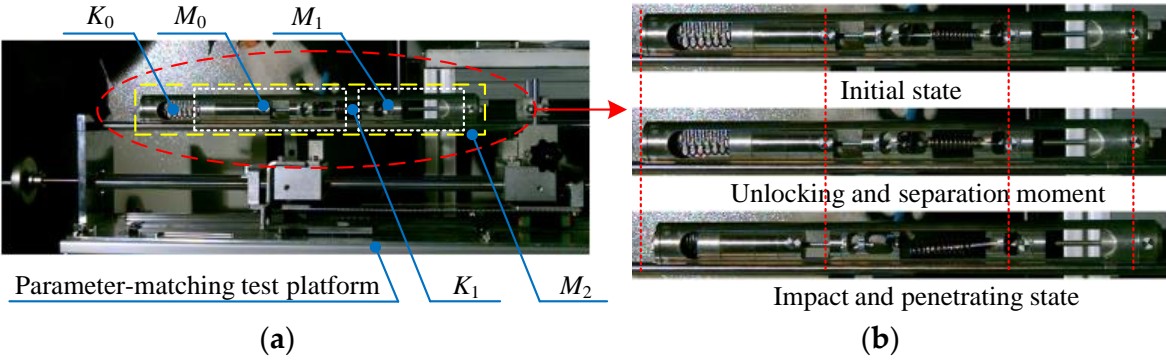

**Figure 7.** The experimental process: (**a**) the preparatory work for the test; (**b**) experimental process of simulated penetrator.

In the testing experiment, to ensure the comparability of experimental results, the compression of the energy storage spring was controlled to be 20 mm each time. During the test, the mass values of key components and the stiffness values of springs were selected successively, and the moving displacement values of the penetrator were obtained under different mass and stiffness combinations. Among all the parameter combinations, 12 groups of matching parameters with ideal effects were chosen to draw out, and the experimental results are shown in the Figure 8.

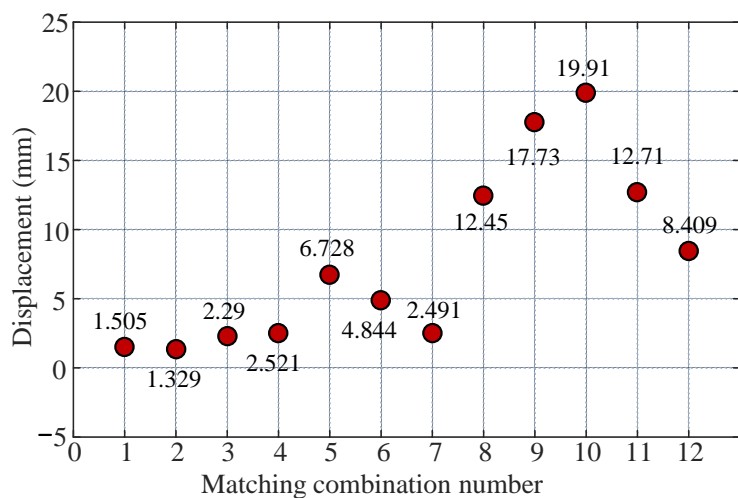

**Figure 8.** The displacement data points under 12 groups of matching parameters.

It can be seen from the figure that the maximum displacement was obtained for mass and stiffness matching combination number 10, which indicated that the parameters of this set of matching combination were optimal among the 12 parameter sets. At this point, the impact actuator mass $M_0 = 801$ g, impact hammer mass $M_1 = 355$ g, penetration unit mass $M_2 = 846$ g, buffer spring stiffness $K_0 = 0.3$ N/mm, and energy storage spring stiffness $K_1 = 9.8$ N/mm. Through the measurement of the high-speed camera, the variation curves of the kinematic displacement and velocity of the penetrator were obtained, as shown in the Figure 9. Under the condition that the energy-storage spring compression was 20 mm, if the 10th group of mass and stiffness parameters was adopted, the maximum velocity and displacement of the penetrator could reach 0.83 m/s and 19.91 mm, respectively, after impact by the impact hammer.

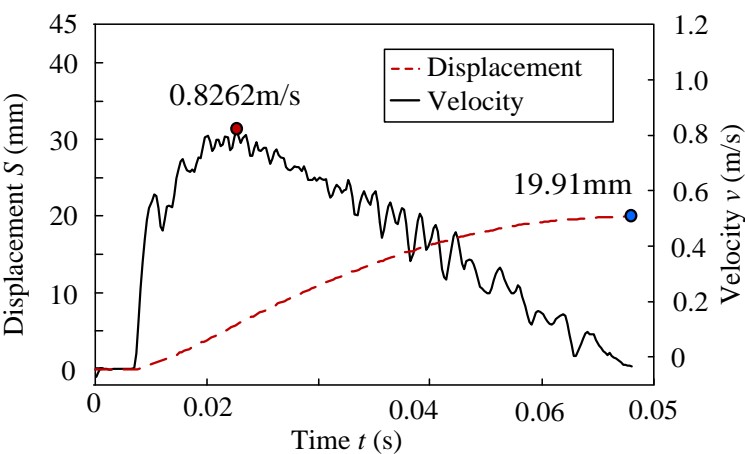

**Figure 9.** The displacement *S* and velocity *v* change with time.

### 4.2. Experiments on the Influence of Dynamic Parameters

The dynamic parameters of the penetrator mainly refer to the impact frequency and impact energy. In previous studies, the configuration parameters of the penetrator were determined by means of theory and simulation [33]. For the given structure parameters of the penetrator, the impact frequency and impact energy had significant effects on the penetrating efficiency. Therefore, in addition to studying the effect of single dynamic parameters on penetrating efficiency, the sensitivity of the two factors to the penetrating efficiency should also be investigated, so as to obtain the dynamic parameter's decision-making method for high penetrating efficiency. This test was carried out on the impact penetrating test platform, which mainly included a driving motor, energy-storage device, penetration prototype, counter weight, lunar soil barrel, high speed camera system, and data acquisition system. Table 8 shows the design specifications of the experiment platform. Figure 10 shows the penetration experiment platform's components and experiment process.

**Table 8.** Design parameters of the penetration test platform.

| Test Platform Envelope Size (mm) | Cone Angle of Penetrator (°) | Drive Motor Power (W) | Working Stroke (mm) | Impact Energy (J) | Impact Frequency (Hz) |
|---|---|---|---|---|---|
| 1100 × 1100 × 1550 | 32.4 | 400 | 0~750 | 0~5 | 0~0.5 |

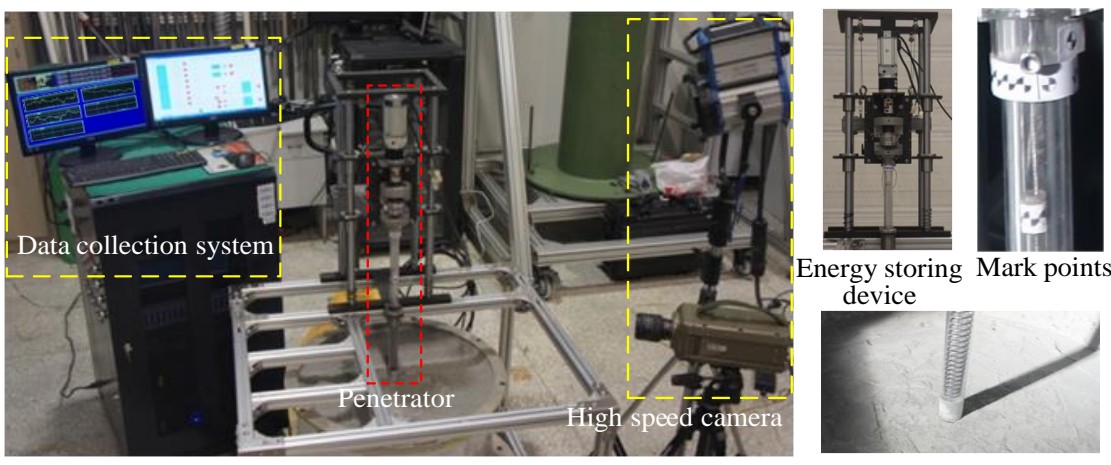

Impact penetrating efficiency test platform          Penetrating process

**Figure 10.** The penetration experiment platform's components and experiment process.

Experimental process: (1) The lunar soil bucket, with a relative compaction of 85%, was transferred to the designated experiment location. (2) A support guider with an envelope size of $\varphi 30$ mm $\times$ 550 mm (Diameter $\times$ Length) and a penetration head with a cone angle of 32.4° were selected, which were fixed on the specified connector. (3) By changing the compression and stiffness of the energy storage spring in the energy storage device, the matching work of impact energy was completed. (4) Mark points used for high-speed camera capture were pasted on impact hammer, support guider and penetrator body in advance, and high-speed camera and supplementary light were adjusted to the best state required in the test; (5) The host computer in the data acquisition system sent instructions to adjust the rotary speed of the driving motor to change the impact frequency. (6) The Mark points were captured by high-speed camera, and the impact energy was calculated by post-processing software. Ultimately, the penetrator could intermittently dive into the lunar soil at the specified impact energy and impact frequency. When the penetration time reached 15 min, the penetrating depth data were recorded and saved.

The experimental results are shown in the Figure 11. It can be seen from the curves that the penetrating depth was positively correlated with the impact frequency or impact energy. As the impact frequency increased, the ramping rate of penetrating depth decreased gradually. The ramping rate of penetrating depth increased with increase of the impact power.

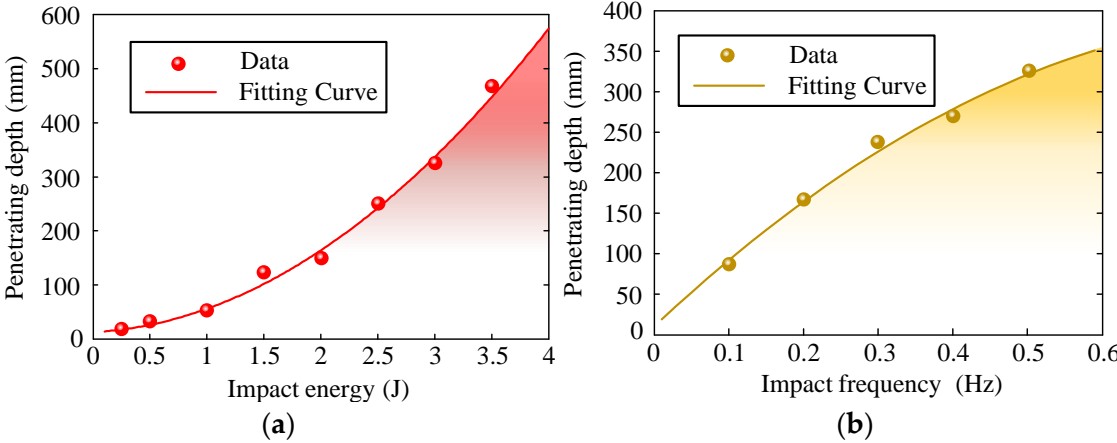

**Figure 11.** The relationship between the penetrating depth and impact energy and impact frequency: (**a**) penetrating depth varies with impact energy when impact frequency is 0.02 Hz; (**b**) penetrating depth varies with impact frequency when the impact energy is 1.5 J.

In the test results, the average penetrating rate was used to characterize the penetrator's penetrating efficiency. The influencing factor's susceptibility of impact energy and impact frequency on penetrating efficiency could be expressed as $F_a$ and $F_b$, and their expressions were presented as follows.

$$
\begin{cases}
F_a = \dfrac{\frac{1}{b}\sum\limits_{i=1}^{a} T_i^2 - \frac{T^2}{ab}}{(a-1)F_T} \\[3mm]
F_b = \dfrac{\frac{1}{a}\sum\limits_{j=1}^{b} T_j^2 - \frac{T^2}{ab}}{(b-1)F_T} \\[3mm]
F_T = \dfrac{\sum\limits_{i=1}^{a}\sum\limits_{j=1}^{b} X_{ij}^2 - \frac{T^2}{ab} - \frac{1}{b}\sum\limits_{i=1}^{a} T_i^2 - \frac{T^2}{ab} - \frac{1}{a}\sum\limits_{j=1}^{b} T_j^2 - \frac{T^2}{ab}}{(a-1)(b-1)}
\end{cases}
\tag{1}
$$

where $T_i$ is the sum of the penetrating efficiency under different impact frequencies for impact energy of group $i$; $T_j$ is the sum of the penetrating efficiency under different impact energies for impact frequency of group $j$; $X_{ij}$ is penetrating efficiency under impact energy in impact energy of group $i$ and impact frequency of group $j$; $a$ and $b$ are the of group numbers

of impact energy and impact frequency set in the test; $T$ is the sum of all the penetrating efficiencies in the orthogonal tests, $T = \sum\limits_{i=1}^{a} \sum\limits_{j=1}^{b} X_{ij} = \sum\limits_{j=1}^{b} \sum\limits_{i=1}^{a} X_{ij}$; $F_T$ is the influence factor susceptibility caused by system errors in orthogonal experiments.

By substituting the test data into Equation (1), $F_a = 13.3$ and $F_b = 9.2$ could be obtained. It was obvious that $F_a > F_b$. According to the two-factor range analysis theory, the sensitivity of impact energy was obviously higher than that of impact frequency in the parameter range of the current penetration schedule. Therefore, the design principle of low-frequency and large impact energy was determined in the parameter selection of the penetration schedule.

### 4.3. Experiments on the Influence of Relative Compactness

The relative compaction of lunar soil simulant was an important index to evaluate the penetrating performance of impact-actuated penetrator. By testing the penetrating efficiency of the penetrator in lunar soil of different relative compaction values, the load bearing capacity of the penetrator was obtained.

This experiment was carried out on the impact penetrating test platform, as shown in the Figure 10. Relevant test parameters were as follows: penetrator head's cone angle was 32.4°; outer diameter of support guider was 30 mm; impact frequency was 0.5 Hz; impact energy was 1 J; and penetrating time was 75 min. The lunar soil simulant with relative compactions of 75%, 80%, 85% and 90% were tested, respectively, and the test results are shown in Figure 12.

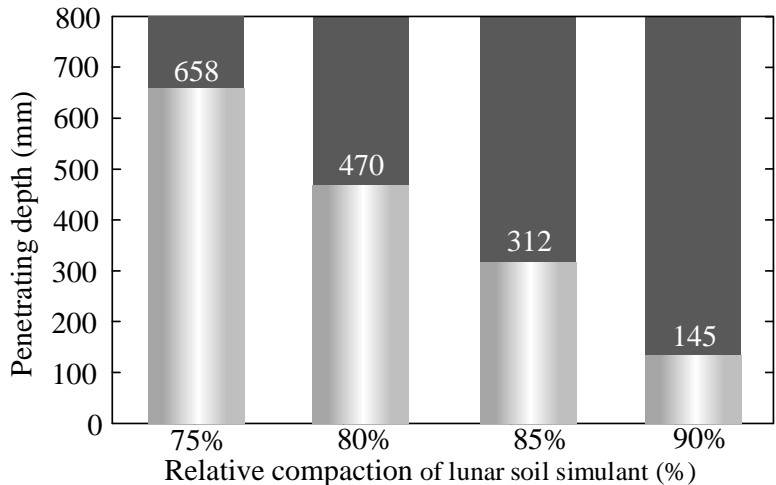

**Figure 12.** The penetrating depths for the penetrator to lunar soil simulant with different relative compaction values.

Experimental results showed that the penetrating depth values of the four kinds of lunar soil simulant with relative compactions of 75%, 80%, 85% and 90% were 658 mm, 470 mm, 312 mm and 145 mm, respectively, under the same experimental conditions. The results revealed that the penetrator could penetrate the simulated lunar soil with a relative compaction of 90%, and the smaller the relative density of the simulated lunar soil, the higher the penetration efficiency of the penetrator. The experimental study could provide guidance for the evaluation of the penetration capacity of the penetrator.

### 4.4. Overall Penetrating Performance of the Optimized Prototype

In order to verify the comprehensive penetrating performance of the optimized impact-actuated penetrator prototype, it was necessary to test the performance of the whole machine for the specified lunar soil simulant. As shown in the Figure 13, the test equipment mainly included the impact actuated penetrator prototype, auxiliary mechanism, data

acquisition system, high-speed camera system, lunar soil simulant and barrel, etc. The size of the impact penetrator used in the experiment was 28.5 mm in diameter and 515 mm in length. The overall mass of the penetrator was 850 g. The energy storage stroke of the penetrator was set to 30 mm, the impact frequency was 0.1 Hz, and the impact energy was 1.22 J. Lunar soil with relative compactness of 85% was adopted in the experiment.

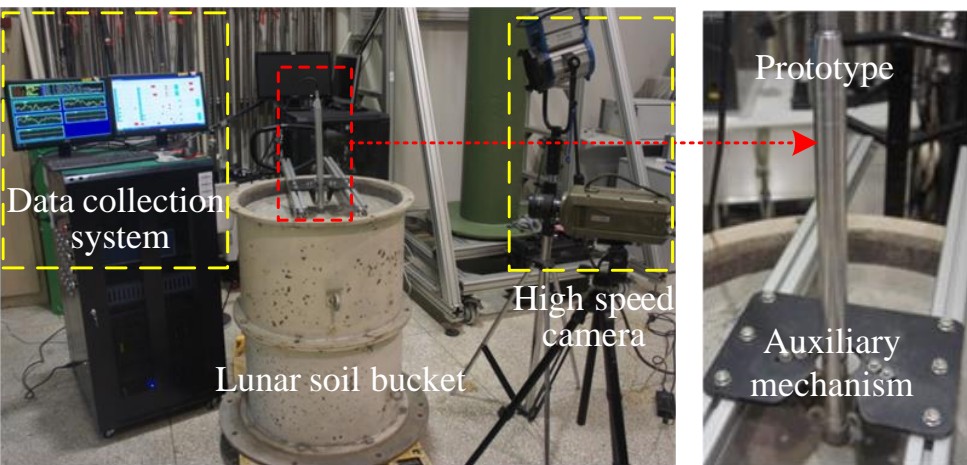

**Figure 13.** The penetration efficiency experiment platform.

Firstly, the lunar soil simulant bucket was transferred to the designated working position. Then, the principal prototype of the impact-actuated penetrator was matched with the interface of the auxiliary mechanism. Mark points for high-speed camera capture were pasted on the penetrator. The high-speed camera and supplementary light were adjusted to the best state required in the test. Finally, the penetrator and data acquisition program were started, and the penetration test was carried out. When the penetrator was fully penetrated into the lunar soil, the experiment was stopped, and the data were recorded and saved. The relation between the penetrating depth and the impact number was acquired in the experiment, as shown in the Figure 14. The curve revealed that the penetration depth was 545 mm after 894 shocks. During the experiment, four different states of the penetrator in penetrating process were captured, which are shown in Figure 14. State (a) presents the initial state in which the nose of the penetrator was in contact with the simulated lunar soil surface; State (b) presents the penetrating state in which the penetrator head was just completely submerged in the simulated lunar soil. State (c) presents the penetrating state in which part of the main body of the penetrator dived into the simulated lunar soil. State (d) presents the state in which the penetrator was completely submerged into the lunar soil simulant. The data revealed that during the first 200 shocks, the penetration depths of the two adjacent shocks had relatively large changes. The average penetration efficiency reached to 2 mm per shock in this period. After that, the penetration depth of single impact decreased gradually. When the penetration depth reached 545 mm, the impact-actuated penetration device still had penetration capability, but the penetration displacement under single impact was very small. This was due to the fact that it is more difficult to extrude deep soil by the penetrator, resulting from the increases of compressive stress of the soil around the penetrator when the penetrator dives completely into the soil.

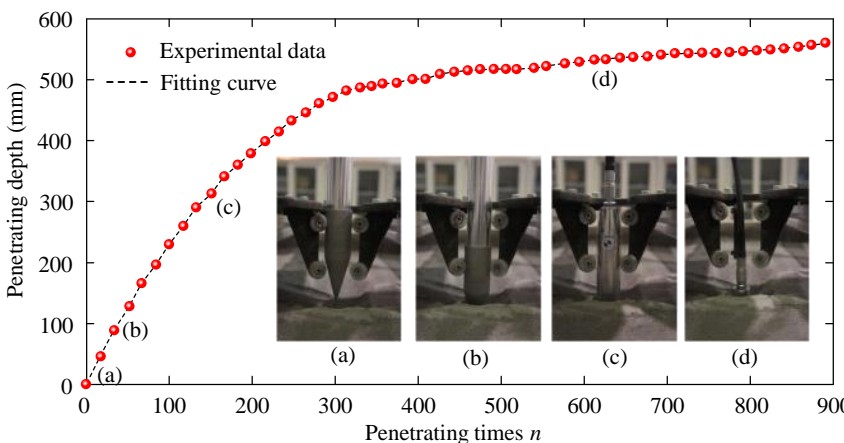

**Figure 14.** The penetrating depths into the lunar soil simulant under different impact numbers.

## 5. Conclusions

After completing the design and optimization of the impact penetrator, it was necessary to conduct experimental research on the penetrating performance of the penetrator in real lunar soil. Due to the lack of a large number of real lunar soil samples, this paper proposed a preparation method of simulated lunar soil samples, based on the analysis and investigation of physical and mechanical parameters of real lunar soil, such as mineral composition, density, porosity ratio, shear and compressibility, etc. The key physical and mechanical parameters of simulated lunar soil samples were close to those of real lunar soil samples. In addition, lunar soil samples with different relative compactness could be obtained by changing the preparation parameters of the lunar soil simulant, which could be used for the penetrator experiments. Based on the simulated lunar soil samples, a large number of testing experiments was carried out. By means of mass and stiffness parameter-matching experiments, the optimal combination of mass and stiffness for the perforator's core components was obtained under specific input conditions. Experiments on the influence of dynamic parameters presented the finding that the sensitivity of impact energy was obviously higher than that of impact frequency in the parameter range of the current penetration schedule, so that the design principle of low-frequency and large impact energy was established. Moreover, the penetrating capability in lunar soil with different compactness was revealed in penetrating experiments with the four kinds of lunar soil simulant with different relative compaction values (75%, 80%, 85%, and 90%). Overall penetrating performance experiments of the optimized penetrator indicated that its penetration depth could reach to 545 mm in lunar soil simulant with a relative compaction of 85% after 894 shocks when impact frequency and the impact energy were 0.1 Hz and 1.22 J, respectively. The preparation of lunar soil simulant and the experimental study of penetrating performance of the impact penetrator provide technical references and support for the development and application of impact penetrators.

**Author Contributions:** Methodology, H.L. and Y.S.; investigation, H.L., Y.L., M.W. and Y.S., writing original draft preparation, H.L. and Y.S.; Writing-review and editing, Y.S. All authors have read and agreed to the published version of the manuscript.

**Funding:** This research was funded by the National Natural Science Foundation of China (Grant No. 51905314), the fellowship of China Postdoctoral Science Foundation (Grant No. 2020M672088), and Natural Science Foundation of Shandong Province (Grant No. ZR2019BEE017).

**Institutional Review Board Statement:** Not applicable.

**Informed Consent Statement:** Not applicable.

**Data Availability Statement:** Not applicable.

**Conflicts of Interest:** The authors declare no conflict of interest.

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
