# Peer review of "Preparation Method of Lunar Soil Simulant and Experimental Verification of the Performance of an Impact Penetrator for Lunar Soil Exploration"

_machines, doi:10.3390/machines10070593_

Round 1
Reviewer 1 Report
Thank you for your study on the performance of a mole architecture.
The trends of the results seem reasonable and provide a useful datapoint.
The architecture appears similar to the MP3 mole. Should it be characterised against that baseline in terms of recoil force, 'second impact', and so on?
In Fig. 11, 'experimental results' are shown. But this looks more like a schematic of the relative effects of energy and frequency. No data points are given. Also, for the energy plot, what is the frequency; and for the frequency plot, what is the energy?
Remember that performance in vacuum may be different for this architecture - Volume 69, Issue 8, 15 April 2022, Pages 3140-3163
Author Response
The authors would like to thank the the reviewer for his or her time and efforts processing this paper and providing invaluable comments and suggestions to enhance the quality of this article. Detailed reply to the reviewer’s comments is listed below. As requested, the article has been revised in the submitted files. We would welcome any additional comments and/or revisions.
Response to Reviewer:
Comment 1: The architecture appears similar to the MP3 mole. Should it be characterised against that baseline in terms of recoil force, 'second impact', and so on?
Authors’ Reply: The impact type penetrator proposed in this article is a compact, lightweight and low power consumption device, using internal hammering mechanism to move downward by compaction of lunar soil. The penetrator after optimized design and parameter-matching in this article has large impact stroke, large energy storage and high impact energy transfer efficiency. The mass and stiffness parameter-matching results show that the recoil force can be absorbed by the buffer spring and the penetration is achieved mainly by the first impact.
Comment 2: In Fig. 11, 'experimental results' are shown. But this looks more like a schematic of the relative effects of energy and frequency. No data points are given. Also, for the energy plot, what is the frequency; and for the frequency plot, what is the energy?
Authors’ Reply: The Figure 11 shows two fitting curves, missing data points. According to the comments of the reviewer, the authors have supplemented the data in the Figure 11. The impact frequency and impact energy are also added in the captions of Figure 11(a) and (b), respectively.
Comment 3: Remember that performance in vacuum may be different for this architecture.
Authors’ Reply: At present, specific experiments to simulate the Moon's gravity environment on earth need high technical requirements and large investment. It is difficult for ground experiments to simulate the actual operating environment of the penetrator on the Moon. Although the experimental results on the earth have some errors compared with the experiment results on the moon, the experiments conducted on the earth is still can provide guidance significance for the development of the penetrator prototype. In the future lunar exploration missions, penetrating experiments for the penetrator developed by the authors may be carried out on the lunar surface to verify the influence of gravity field on the its penetrating performance in real lunar soil.

Reviewer 2 Report
The methods and the results presented are interesting to me, but the form jeopardizes the contents. A revision of english is needed. I underlined some issues in the attached pdf.

Author Response
The authors would like to thank the reviewer for his or her time and efforts processing this paper and providing invaluable comments and suggestions to enhance the quality of this article. Detailed reply to the reviewer’s comments is listed below. As requested, the article has been revised in the submitted files. We would welcome any additional comments and/or revisions. Finally, the authors have checked the paper against the attached Editorial Checklist, and the issues have been addressed in the amended version.
Comments from Reviewer:
The methods and the results presented are interesting to me, but the form jeopardizes the contents. A revision of English is needed. I underlined some issues in the attached pdf.
Response to Reviewer:
The authors have carefully revised the format and language according to the comments and suggestions of the reviewer. These changes have been highlighted in the revised version. As for the specific questions mentioned in the article by the reviewer, the authors' answers are added in the article, which are also highlighted in the revised version.

Round 2
Reviewer 1 Report
Thank you for the edits, I hope you agree that they are useful.